# Distance effect of single atoms on stability of cobalt oxide catalysts for acidic oxygen evolution

Zhirong Zhang [1,5], Chuanyi Jia[2,5], Peiyu Ma [3,5], Chen Feng[1], Jin Yang[1], Junming Huang[1], Jiana Zheng[1], Ming Zuo[1], Mingkai Liu[4], Shiming Zhou [1] ✉ & Jie Zeng [1,4] ✉

Developing efficient and economical electrocatalysts for acidic oxygen evolution reaction (OER) is essential for proton exchange membrane water electrolyzers (PEMWE). Cobalt oxides are considered promising non-precious OER catalysts due to their high activities. However, the severe dissolution of Co atoms in acid media leads to the collapse of crystal structure, which impedes their application in PEMWE. Here, we report that introducing acid-resistant Ir single atoms into the lattice of spinel cobalt oxides can significantly suppress the Co dissolution and keep them highly stable during the acidic OER process. Combining theoretical and experimental studies, we reveal that the stabilizing effect induced by Ir heteroatoms exhibits a strong dependence on the distance of adjacent Ir single atoms, where the OER stability of cobalt oxides continuously improves with decreasing the distance. When the distance reduces to about 0.6 nm, the spinel cobalt oxides present no obvious degradation over a 60-h stability test for acidic OER, suggesting potential for practical applications.

Proton exchange membrane water electrolysis (PEMWE) driven by renewable electricity is the most promising route to the production of clean hydrogen fuels[1,2]. The large-scale deployment of PEMWE devices is predominantly obstructed by the efficient catalysts for oxygen evolution reaction (OER) in acidic media. Currently, precious metal oxide $IrO_2$ is generally considered to be the most stable electrocatalyst for the acidic OER[3,4]. However, the serious scarcity of iridium, with global production of only about 7 tons per year, makes it difficult for PEMWE to reach sustainable terawatt level goals, where more than 40 years of annual iridium production are estimated to be required[5,6]. Great efforts have been devoted to exploring effective strategies to decrease Ir

loadings in acidic OER catalysts, such as constructing Ir-metal clusters[7,8], heterostructures[9–11], and deposition of Ir species on suitable support[12,13]. However, these strategies still utilize relatively abundant Ir species. Therefore, it is highly desired but remains a major challenge, to develop efficient and durable catalysts with low Ir consumption for acidic OER.

Cobalt oxides are regarded as a very promising candidate for catalyzing OER due to their earth abundance and excellent activity[14–17]. In particular, spinel cobalt oxides attract extensive attention benefiting from their flexible composition and tunable structure[18,19]. Nevertheless, these oxides are only stable during the OER process in neutral or alkaline media. Under acidic conditions, the dissolution of cobalt

[1]Hefei National Research Center for Physical Sciences at the Microscale, Key Laboratory of Strongly-Coupled Quantum Matter Physics of Chinese Academy of Sciences, Key Laboratory of Surface and Interface Chemistry and Energy Catalysis of Anhui Higher Education Institutes, Department of Chemical Physics, University of Science and Technology of China, Hefei, Anhui 230026, PR China. [2]Guizhou Provincial Key Laboratory of Computational Nano-Material Science, Institute of Applied Physics, Guizhou Education University, Guiyang, Guizhou 550018, PR China. [3]National Synchrotron Radiation Laboratory, Key Laboratory of Precision and Intelligent Chemistry, iChEM (Collaborative Innovation Center of Chemistry for Energy Materials), University of Science and Technology of China, Hefei, Anhui 230026, PR China. [4]School of Chemistry & Chemical Engineering, Anhui University of Technology, Ma'anshan, Anhui 243002, PR China. [5]These authors contributed equally: Zhirong Zhang, Chuanyi Jia, Peiyu Ma. ✉e-mail: zhousm@ustc.edu.cn; zengj@ustc.edu.cn

atoms leads to the collapse of crystal structure, which limited their application in PEMWE[20–22]. Recently, some pioneering works have reported that the introduction of acid-resistant heteroatoms such as Pb, Mn, and Sb into cobalt oxides could increase the stability of these OER catalysts in acid[6,18,23,24]. It was proposed that the heteroatoms would strengthen metal-oxygen bindings[6], activate the self-healing process[23], or stabilize the lattice[24]. However, atomic-level insight into the stabilizing effect induced by the acid-resistant heteroatoms is still lacking, which is crucial for designing highly efficient earth-abundant catalysts for acidic OER.

Herein, combining theoretical and experimental studies on single-atom Ir-introduced spinel cobalt oxides, we provided an in-depth understanding of the heteroatom's role on OER stability at the atomic level. Our density functional theory (DFT) calculations revealed that the introduction of Ir single atoms can significantly increase the migration energy of the nearest Co atoms, whereas has less influence on the distant ones. Accordingly, we further studied the inter-site distance effect of adjacent Ir atoms, i.e., Ir-Ir distance, on the stability of Co atoms between them. We found that the stability of the in-between Co atom was continuously enhanced until the Ir-Ir distance was lowered down to the limited level of 0.56 nm where the Co atom was sandwiched by two Ir atoms. Experimentally, we successfully synthesized a series of single-atom catalysts with different Ir-Ir distances by adjusting the density of Ir single atoms in spinel cobalt oxides. Electrochemical measurements further demonstrated that the stabilizing effect induced by Ir single atoms was closely related to the Ir-Ir distance. With the decrease of the Ir-Ir distance, the electrochemical durability of these catalysts in acid exhibited a gradual improvement, accompanied by a significant decrease in the dissolution of Co species, which well confirmed the theoretical results. When the Ir-Ir distance reached about 0.6 nm, the long-term durability test at 10 mA cm$_{geo}^{-2}$ revealed that the spinel cobalt oxide remained stable under pH = 1 with just a small increase in the potential of about 20 mV

after a 60 h continuous operation. Our work offered essential guidance for the precise design of highly stable earth-abundant OER catalysts in acid media.

## Results
### Exploration of the distance-related stabilizing effect
We began with DFT calculations to study how the introduction of acid-resistant heteroatoms influences the stability of Co atoms in spinel oxides. Since the dissolution of cobalt oxides in acid involves the migration of Co atoms at the surface[6,25], we calculated the migration energies of these lattice atoms to evaluate their stabilities. Taking spinel Cu$_{0.3}$Co$_{2.7}$O$_4$ as a model (Fig. 1a), we proposed a migrating process of the Co atom at octahedral sites as shown in Fig. 1b, where the Co atom moves away from the (110) face via a transition state with an energy barrier of 1.58 eV. After Ir single atoms were introduced into the octahedral sites of spinel oxide (Fig. 1c), we first calculated the migration energy of the nearest Co atom (labeled as A) to the Ir single atom. We found that the energy increased to 1.70 eV (Supplementary Fig. 1a), indicating that the introduction of the Ir single atom can significantly enhance the stability of the nearest Co atom. Subsequently, we further assessed the influence of Ir single atom on the migration of distant Co atoms, i.e., the next nearest Co atom as labeled B in Fig. 1c. The calculated migration energy of B-site Co was 1.63 eV (Supplementary Fig. 1b), which suggested that the B-site Co atom was less stable than the A-site Co but more than the Co atom without introducing Ir (Fig. 1d). In other words, the stabilizing effect induced by Ir single atoms became weaker as the distance between Ir and Co atoms increased. This result strongly suggested that it is necessary to control the distance of adjacent Ir single atoms by adjusting the density of introduced Ir single atoms for improving the stability of the cobalt oxide catalysts.

To further explore the distance-dependent stabilizing effect of Ir single atoms, we constructed various structural models of Ir atomically

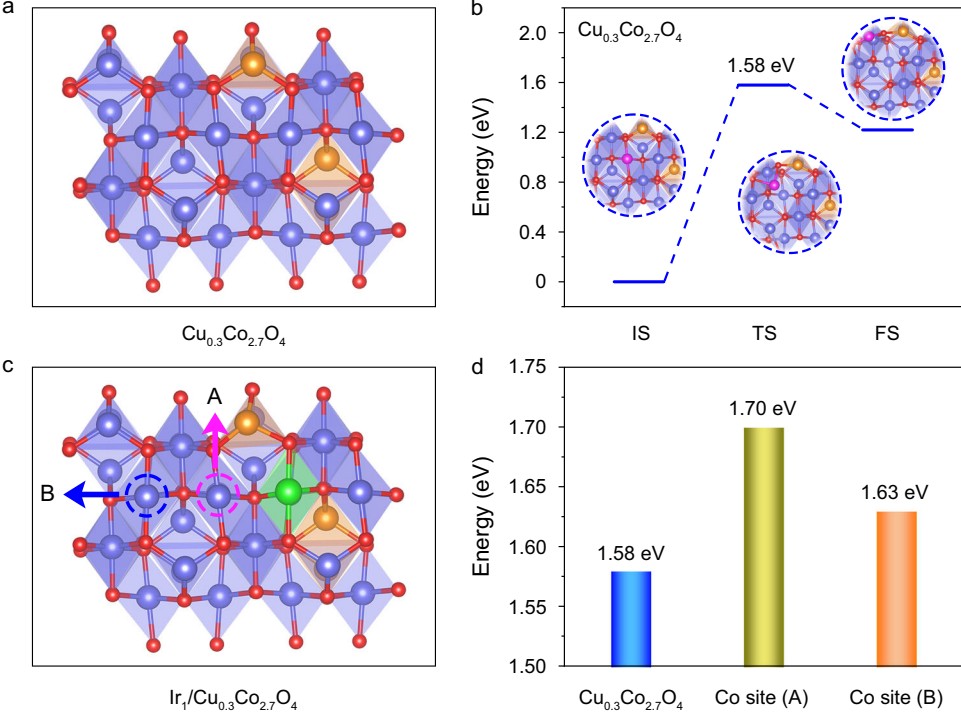

**Fig. 1 | Optimized structural models and calculated migration energies of Cu$_{0.3}$Co$_{2.7}$O$_4$ and Ir$_1$/Cu$_{0.3}$Co$_{2.7}$O$_4$. a** Optimized structural model of Cu$_{0.3}$Co$_{2.7}$O$_4$. Red, blue, and brown spheres represent O, Co, and Cu atoms, respectively. **b** Calculated migration energies of Co atoms on Cu$_{0.3}$Co$_{2.7}$O$_4$. Pink spheres represent migrated Co atoms. **c** Optimized structural model of Ir$_1$/Cu$_{0.3}$Co$_{2.7}$O$_4$. Green spheres represent the Ir atom. Pink and blue circles represented the Co site located at different distances from the Ir atom. **d** Calculated migration energies of Co atoms at different sites.

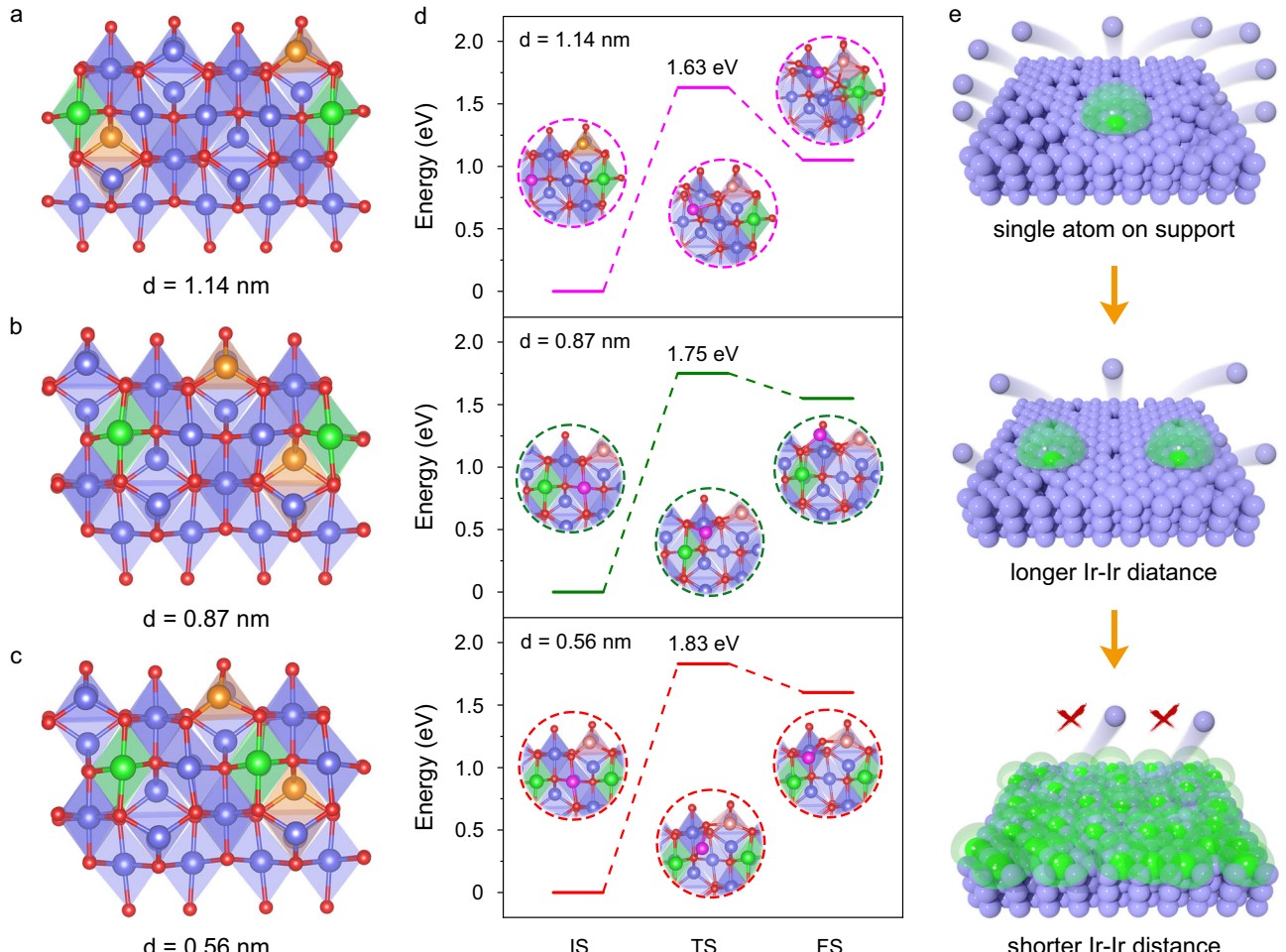

**Fig. 2 | Optimized structural models and calculated migration energies of Ir₁/Cu₀.₃Co₂.₇O₄ with different Ir-Ir distances.** Optimized structural models of Ir₁/Cu₀.₃Co₂.₇O₄ with different Ir-Ir distances. Ir₁/Cu₀.₃Co₂.₇O₄ with $d$ = 1.14 nm (**a**), $d$ = 0.87 nm (**b**), and $d$ = 0.56 nm (**c**). Red, blue, brown, and green spheres represent O, Co, Cu, and Ir atoms, respectively. **d** Calculated migration energies of Co atoms on Ir₁/Cu₀.₃Co₂.₇O₄ with different Ir-Ir distances. The inset structures represent the initial state (IS), transition state (TS), and final state (FS), respectively. Pink spheres represent migrated Co atoms. **e** Schematic illustration of the distance effect of Ir single atoms on the stability of cobalt oxide catalysts. Blue and green spheres represent Co atoms and Ir single atoms.

doped spinel cobalt oxides, Ir₁/Cu₀.₃Co₂.₇O₄, with different Ir-Ir distances (d). Figure 2a–c showed the cases with $d$ = 1.14, 0.87, and 0.56 nm, in which the number of Co atoms between two adjacent Ir atoms was 3, 2, and 1, respectively. For $d$ = 1.14 nm, the calculated migration energy of the centered Co atom was 1.63 eV. For $d$ = 0.87 nm, the energy was 1.75 eV (Fig. 2d). Both the energies are essentially the same as those for B-site and A-site Co atoms in Fig. 1c, respectively. For $d$ = 0.56 nm, it took 1.83 eV to enable the migration of the Co atom sandwiched by two Ir single atoms (Fig. 2d). The increased migration energies indicated that the shorter the distance between adjacent Ir single atoms, the more stable the Co atoms on the catalyst surface. The above results indicated that Ir single atoms can stabilize their neighboring lattice but have limited effect on distant ones. When the distance between adjacent Ir single atoms was too large, the stabilizing effect of Ir single atoms was localized. Once the distance between Ir single atoms was reduced to a specific value, the stabilizing effect would cover the spinel oxides effectively, thus stabilizing the entire spinel oxides under acidic conditions (Fig. 2e).

### Identifying Ir₁/Cu₀.₃Co₂.₇O₄ with different Ir single atoms distance

Inspired by the theoretical calculations, we synthesized spinel oxide Cu₀.₃Co₂.₇O₄ and a series of Ir₁/Cu₀.₃Co₂.₇O₄ with different Ir-Ir distances through a high-temperature pyrolysis method (see Methods).

The distance between Ir single atoms was modulated by adjusting the density of Ir single atoms on Cu₀.₃Co₂.₇O₄. Transmission electron microscopy (TEM) images showed that the as-prepared Cu₀.₃Co₂.₇O₄ and Ir₁/Cu₀.₃Co₂.₇O₄ with different Ir-Ir distances presented similar morphologies (Supplementary Fig. 2). X-ray diffraction (XRD) patterns revealed that all the samples were in the spinel-type structures with a F$d$-3m space group (Supplementary Fig. 3). Moreover, the Raman spectra displayed four characteristic peaks located at 190, 470, 510, and 682 cm⁻¹ for these oxides, which were assigned to F₂g, Eg, F₂g, and A₁g vibration originated from the spinel lattice, respectively (Supplementary Fig. 4)[18,26]. These results demonstrated that the introduction of Ir heteroatoms into the spinel cobalt oxides not only formed no detectable impurity phase but also unchanged the spinel structure.

Figure 3a–c showed the aberration-corrected high-angle annular dark-field scanning TEM (HAADF-STEM) images for three Ir₁/Cu₀.₃Co₂.₇O₄ samples with different Ir-Ir distances. Individual bright spots in contrast to the Cu₀.₃Co₂.₇O₄ were found, which indicated that the Ir species were atomically dispersed in the spinel cobalt oxides. Moreover, by averaging the Ir-Ir distances of more than two hundred Ir-Ir pairs in the HAADF-STEM images, the value of d was estimated to be about 1.1, 0.8, and 0.6 nm, respectively (Fig. 3d–i). Energy dispersive X-ray (EDX) elemental mapping images revealed that the Ir atoms were uniformly distributed in all three Ir₁/Cu₀.₃Co₂.₇O₄ samples

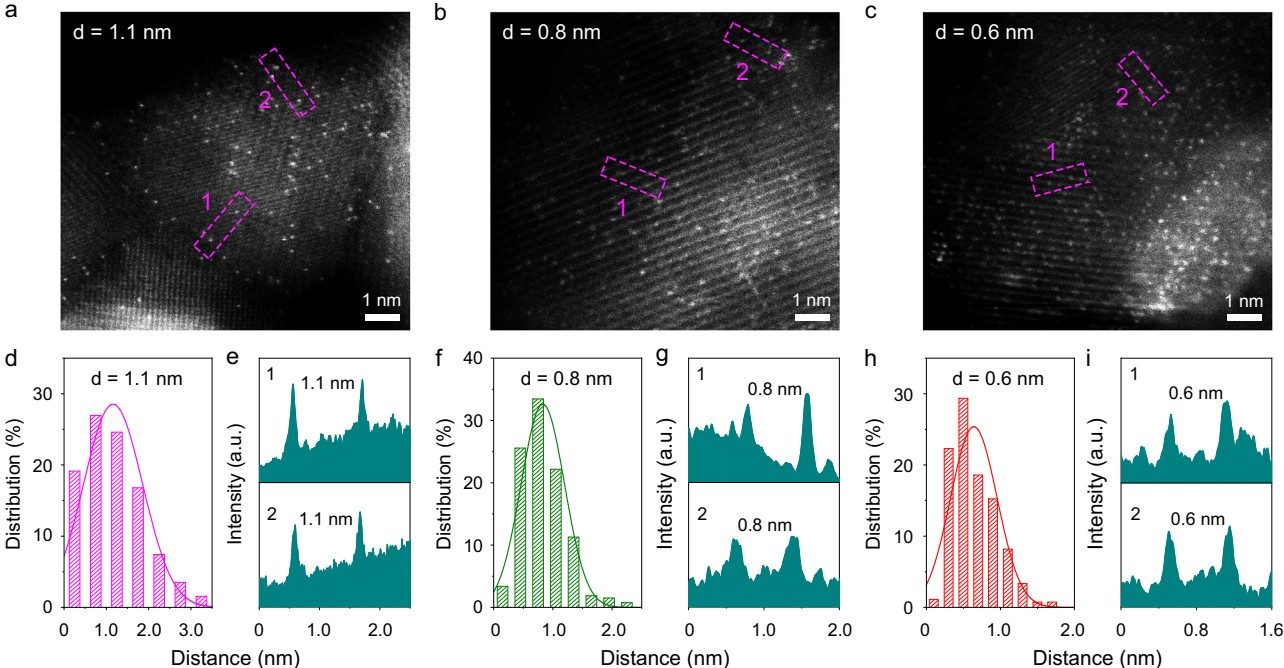

**Fig. 3 | Structural characterizations of $Ir_1/Cu_{0.3}Co_{2.7}O_4$ with different Ir-Ir distances.** HAADF-STEM images of $Ir_1/Cu_{0.3}Co_{2.7}O_4$ with different Ir-Ir distances. $Ir_1/Cu_{0.3}Co_{2.7}O_4$ with $d = 1.1$ nm (**a**), $d = 0.8$ nm (**b**), and $d = 0.6$ nm (**c**). Distance distribution of Ir single atoms (**d**) and intensity profile (**e**) of atoms located at the square frame in (**a**). Distance distribution of Ir single atoms (**f**) and intensity profile (**g**) of atoms located at the square frame in (**b**). Distance distribution of Ir single atoms (**h**) and intensity profile (**i**) of atoms located at the square frame in (**c**).

(Supplementary Fig. 5). The quantitative analysis by inductively coupled plasma-atomic emission spectrometry (ICP-AES) and inductively coupled plasma-mass spectrometry (ICP-MS) showed that the contents of Ir elements were 1.2, 2.1, and 3.6 wt% for $Ir_1/Cu_{0.3}Co_{2.7}O_4$ with $d = 1.1$, 0.8, and 0.6 nm, respectively.

The electronic structure and coordination environment of $Ir_1/Cu_{0.3}Co_{2.7}O_4$ with different Ir-Ir distances were further characterized by X-ray absorption near-edge spectroscopy (XANES) and extended X-ray absorption fine structure (EXAFS) spectroscopy. The Co $K$-edge XANES spectra showed that the absorption edges of $Ir_1/Cu_{0.3}Co_{2.7}O_4$ with different Ir-Ir distances overlapped with that of $Cu_{0.3}Co_{2.7}O_4$, suggesting similar valence states for Co species in all the oxides (Fig. 4a)[27,28]. The Co $K$-edge EXAFS spectra presented three similar characteristic peaks at about 1.4, 2.3, and 3.0 Å, corresponding to Co-O, $Co_{oct}$-$Co_{oct}$ (octahedral site), and $Co_{tet}$-$Co_{tet}$ (tetrahedral site) (Fig. 4b)[29,30], which implies that the coordination environment of Co sites exhibited no obvious change after the introduction of Ir single atoms. Moreover, the Co and Cu $L$-edge X-ray absorption spectroscopy (XAS) also revealed similar valence states of Co and Cu ions in these as-prepared samples (Supplementary Fig. 6a)[31–35]. The above results indicated that no visible changes in the crystal and electronic structures of the cobalt oxides were found after the introduction of Ir single atoms with different distances. Figure 4c shows the Ir $L_3$-edge XANES spectra of $Ir_1/Cu_{0.3}Co_{2.7}O_4$ with different Ir-Ir distances, where $IrO_2$ and Ir foil were used as references. The intensity of the white line was found to decline gradually with decreasing the Ir-Ir distances, which suggested a decrease in the valence state of Ir species[36,37]. The Ir $L_3$-edge EXAFS spectra exhibited two characteristic peaks at about 2.0 Å and 3.0 Å, which were ascribed to first-shell Ir-O coordination and second-shell Ir-Co coordination, respectively (Fig. 4d)[38,39]. By fitting the experimental EXAFS spectra, the Ir-O and Ir-Co coordination numbers of $Ir_1/Cu_{0.3}Co_{2.7}O_4$ with different Ir-Ir distances were determined to be about six and three, respectively (Supplementary Fig. 7 and Table 1). The fitting results confirmed that the Ir single atoms were incorporated into the octahedral sites of $Cu_{0.3}Co_{2.7}O_4$.

## Electrocatalytic evaluation towards acidic oxygen evolution

To evaluate the stabilizing effect of Ir single atoms with different distances on $Cu_{0.3}Co_{2.7}O_4$, we recorded the polarization curves in a standard three-electrode system under acidic media. For the pristine $Cu_{0.3}Co_{2.7}O_4$, the current density showed an obvious decrease as the number of scans increased (Fig. 5a). After 1000 scan cycles, the current density decreased by 87.5%. As Ir single atoms with $d = 1.1$ and 0.8 nm were introduced into $Cu_{0.3}Co_{2.7}O_4$, the current density decreased by 76.3% and 44.2% after 1000 scan cycles, respectively (Fig. 5a, b). When the Ir-Ir distance was further reduced to 0.6 nm, the current density showed an inconspicuous decrease during the sequential scans, indicating excellent OER stability for this sample in acidic media (Fig. 5a, b). The dissolution of Co species under different scan cycles was also measured to explore the stability of $Cu_{0.3}Co_{2.7}O_4$ and $Ir_1/Cu_{0.3}Co_{2.7}O_4$ with different Ir-Ir distances. For Co species in $Cu_{0.3}Co_{2.7}O_4$, they were gradually dissolved with increasing the number of scan cycles. Specifically, 76.7% of Co species were dissolved after 1000 scan cycles (Fig. 5c). When Ir = 1.1 and 0.8 nm were introduced into the $Cu_{0.3}Co_{2.7}O_4$, the dissolution rate of Co species slowed down, indicating the Ir single atoms could stabilize the $Cu_{0.3}Co_{2.7}O_4$. When Ir = 0.6 nm were introduced into the $Cu_{0.3}Co_{2.7}O_4$, Co species were merely dissolved (just 3.6%) after 1000 scan cycles, indicating high stability of $Cu_{0.3}Co_{2.7}O_4$ with $d = 0.6$ nm. For Ir species, the results demonstrated that as the Ir-Ir distance decreases, the dissolution rate of Ir species on the catalyst surface slows down (Supplementary Fig. 8). Notably, the Ir species in $Ir_1/Cu_{0.3}Co_{2.7}O_4$ with $d = 0.6$ nm were just dissolved 2.1% after 1000 scan cycles, indicating the high stability of this sample during acidic OER condition. Long-term chronopotentiometry at a constant current density of 10 mA cm$^{-2}_{geo}$ was also carried out to estimate the durability of $Cu_{0.3}Co_{2.7}O_4$ and $Ir_1/Cu_{0.3}Co_{2.7}O_4$ with different Ir-Ir distances. As shown in Fig. 5d, the durability of these catalysts was improved as the Ir-Ir distance decreased. For $d = 0.6$ nm, the catalyst remained stable with just a small increase in the potential of about 20 mV after a 60 h continuous operation. Notably, the $Ir_1/Cu_{0.3}Co_{2.7}O_4$ with $d = 0.6$ nm exhibited a better stability in comparison

to that of commercial $IrO_2$. The above results demonstrated that the stabilizing effect induced by Ir single atoms was strongly dependent on the distance of adjacent single atoms. When Ir single atoms with $d = 0.6$ nm were introduced into the $Cu_{0.3}Co_{2.7}O_4$, the stabilizing effect was superimposed on each other, thus stabilizing the entire catalysts.

To exclude the possible influence of carbon corrosion and gas bubbles on the OER stability during the above electrochemical measurements, we further recorded the polarization curves in a three-electrode system with the catalysts loading on Ti felt (Supplementary Figs. 9 and 10) and in a flow-cell setup (Supplementary Fig. 11). Similar improvement in the OER stability by decreasing the Ir-Ir distance can be found for both measurements, which indicated that the distance effect of Ir single atoms on the stability of cobalt oxide catalysts is intrinsic. Moreover, considering the difference in the content of Ir species on the electrode may influence the stability of the catalysts, we also carried out the electrochemical evaluation for these catalysts by fixing the Ir loadings on the electrode at $25\,\mu g\,cm^{-2}$. For the $Ir_1/Cu_{0.3}Co_{2.7}O_4$ with $d = 1.1$ and 0.8 nm, the current density showed an obvious decrease as the number of scans increased (Supplementary Fig. 12a, b, d). For the $Ir_1/Cu_{0.3}Co_{2.7}O_4$ with $d = 0.6$ nm, the current density showed a negligible decrease during the sequential scans, indicating excellent OER stability for this sample in acidic media (Supplementary Fig. 12c, d). The above results further proved that the stabilizing effect induced by Ir single atoms was only related to the distance of adjacent Ir single atoms.

The excellent stability of $Ir_1/Cu_{0.3}Co_{2.7}O_4$ with $d = 0.6$ nm was also supported by the HAADF-STEM image, EDX elemental mapping images, and XAS after the durability test. As shown in the HAADF-STEM image, Ir single atoms preserved their isolated dispersion on the $Cu_{0.3}Co_{2.7}O_4$ (Supplementary Fig. 13a). The EDX elemental mapping image showed that Ir species were still evenly distributed across the

catalyst (Supplementary Fig. 13b). The Co $L_3$- and $L_2$-edge peaks displayed negligible change compared with that before durability test, suggesting an unchanged valence state of Co (Supplementary Fig. 13c). In addition, the O $K$-edge XAS also showed ignorable changes in the characteristic peaks, indicating a stable structure of the $Ir_1/Cu_{0.3}Co_{2.7}O_4$ with $d = 0.6$ nm during the durability test (Supplementary Fig. 13d).

To further prove the distance effect of Ir single atoms on the stability of $Cu_{0.3}Co_{2.7}O_4$ during acidic OER, in-situ XAFS were performed (Supplementary Fig. 14a, b). As shown in Supplementary Fig. 14c, the absorption edge of Co $K$-edge XANES spectra for $Cu_{0.3}Co_{2.7}O_4$ exhibited inconspicuous changes with increasing voltage from open circle potential (OCP) to 1.7 V. However, the EXAFS spectra revealed that the intensity of the two main peaks from the Co-O and $Co_{oct}$-$Co_{oct}$ coordination exhibited a decreasing tendency with increasing voltage, indicating $Cu_{0.3}Co_{2.7}O_4$ was dissolved during acidic OER (Supplementary Fig. 14d). For comparison, in-situ XAFS at the Co $K$-edge of $Ir_1/Cu_{0.3}Co_{2.7}O_4$ with $d = 0.6$ nm was also conducted. The absorption edge of Co $K$-edge XANES spectra for $Ir_1/Cu_{0.3}Co_{2.7}O_4$ with $d = 0.6$ nm exhibited negligible changes with increasing voltage from OCP to 1.7 V, suggesting the excellent stability of the catalysts under oxidative potentials[40] (Supplementary Fig. 14e). Meanwhile, the EXAFS spectra revealed that the position and intensity of the three main peaks from the Co-O, $Co_{oct}$-$Co_{oct}$, and $Co_{tet}$-$Co_{tet}$ coordination exhibited an insignificant change with increasing voltage, indicating a stable structure of this sample during acidic OER (Supplementary Fig. 14f). These in-situ XAFS results demonstrated that Ir single atoms with $d = 0.6$ nm effectively stabilized the structure of $Cu_{0.3}Co_{2.7}O_4$ during acidic OER. Considering the Ir single atoms were introduced into the lattice of $Cu_{0.3}Co_{2.7}O_4$, the stabilizing effect may originated from the formation of the Ir-O-

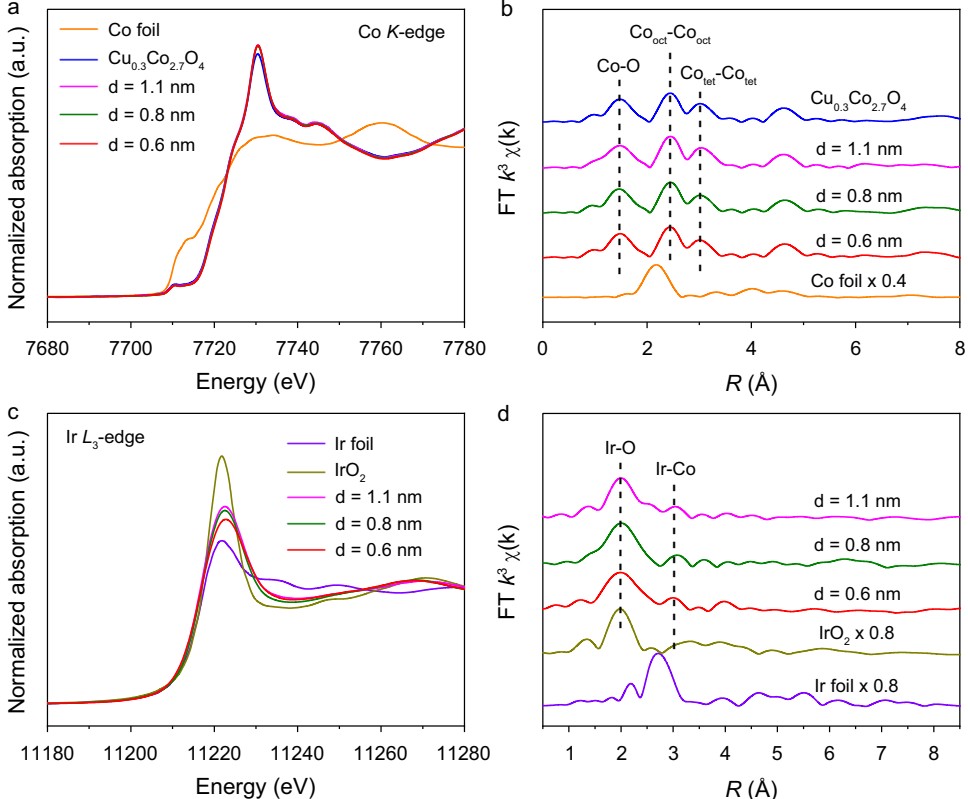

**Fig. 4 | Electronic structure characterizations of $Ir_1/Cu_{0.3}Co_{2.7}O_4$ with different Ir-Ir distances.** Normalized XANES (**a**) and EXAFS (**b**) spectra at the Co $K$-edge of $Cu_{0.3}Co_{2.7}O_4$ and $Ir_1/Cu_{0.3}Co_{2.7}O_4$ with different Ir-Ir distances. Co foil was used as a reference. Normalized XANES (**c**) and EXAFS spectra (**d**) at the Ir $L_3$-edge of $Ir_1/Cu_{0.3}Co_{2.7}O_4$ with different Ir-Ir distances. Ir foil and $IrO_2$ were used as references.

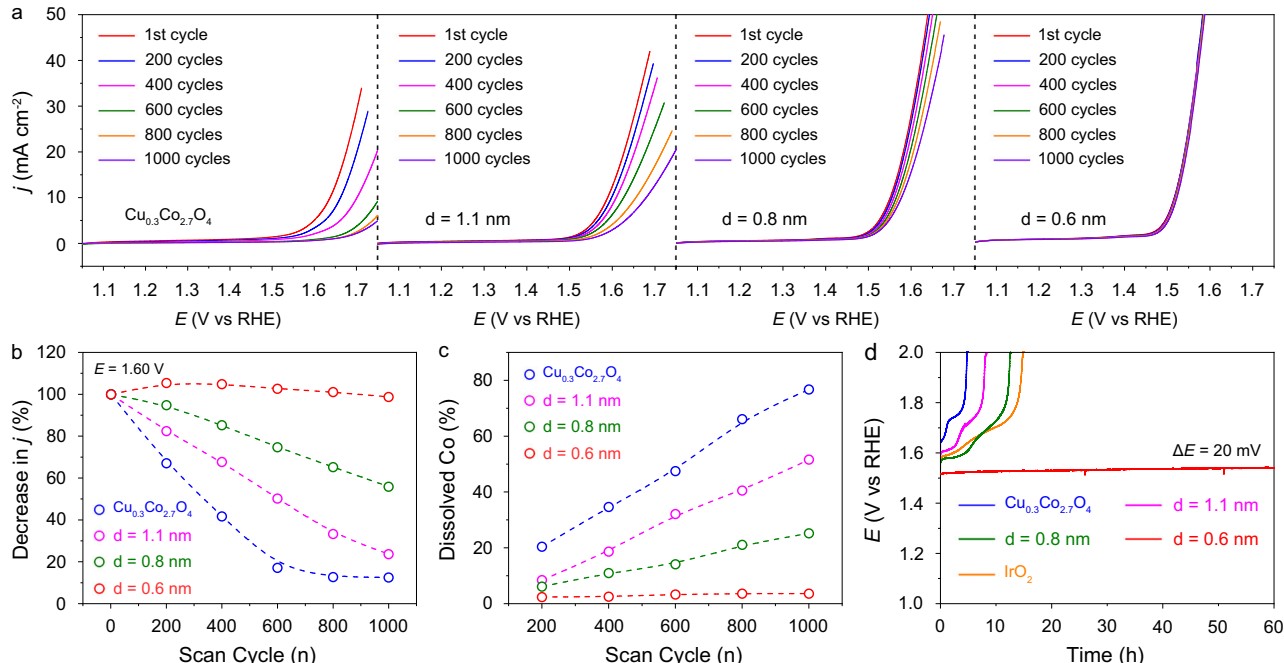

**Fig. 5 | Stability evaluation towards acidic oxygen evolution. a** Polarization curves of catalysts at different scan cycles in 0.1 M HClO₄ electrolyte. The displayed polarization curves are the 1, 200, 400, 600, 800, and 1000 cycles, respectively. **b** Decrease in current densities under different scan cycles of Cu₀.₃Co₂.₇O₄ and Ir₁/Cu₀.₃Co₂.₇O₄ with different Ir-Ir distances. The selected overpotential was 1.60 V

($E$ vs RHE) for all catalysts. **c** Dissolution of Co species under different scan cycles of Cu₀.₃Co₂.₇O₄ and Ir₁/Cu₀.₃Co₂.₇O₄ with different Ir-Ir distances. **d** Chronopotentiometry curves of Cu₀.₃Co₂.₇O₄, Ir₁/Cu₀.₃Co₂.₇O₄ with different Ir-Ir distances, and commercial IrO₂ towards acidic OER at 10 mA cm⁻².

Co structure. As the distance of adjacent Ir single atoms narrowed to a suitable value, the Ir-O-Co structure covered the entire Cu₀.₃Co₂.₇O₄, making the stabilizing effect superimposed on each other, thus significantly stabilizing the entire spinel oxides.

The electrocatalytic activity of Ir₁/Cu₀.₃Co₂.₇O₄ with different Ir-Ir distances towards acidic OER was also evaluated. For comparison, Cu₀.₃Co₂.₇O₄ was measured under the same conditions. Supplementary Fig. 15a displayed that the introduction of Ir single atoms improves the OER activity. Moreover, the current density of Ir₁/Cu₀.₃Co₂.₇O₄ increased with decreasing the distance of adjacent Ir single atoms. Specifically, Ir₁/Cu₀.₃Co₂.₇O₄ with $d = 0.6$ nm only required an overpotential of 290 mV to deliver a current density of 10 mA cm⁻², which was 120, 70, and 30 mV lower than those of Cu₀.₃Co₂.₇O₄, Ir₁/Cu₀.₃Co₂.₇O₄ with $d = 1.1$ and 0.8 nm, respectively (Supplementary Fig. 15b). The results indicated that the reduction of the distance between Ir single atoms significantly enhanced the catalytic activity under the acidic condition. To elucidate the reaction kinetics, we analyzed the Tafel slopes of these catalysts, where the Tafel slope of Ir₁/Cu₀.₃Co₂.₇O₄ with $d = 0.6$ nm gave the lowest value of 71 mV dec⁻¹ (Supplementary Fig. 16). This suggested its fastest kinetics among this catalysts[41]. The reaction kinetics was further reflected by the electrochemical impedance spectroscopy (EIS) measurements. A remarkable decrease of charge transfer resistance ($R_{ct}$) was found in the Ir₁/Cu₀.₃Co₂.₇O₄ with $d = 0.6$ nm compared with other catalysts, confirming its fastest charge transfer (Supplementary Fig. 17)[42,43]. The improved catalytic performance and accelerated reaction kinetics of Ir₁/Cu₀.₃Co₂.₇O₄ with $d = 0.6$ nm may be attributed to the optimized electronic structure of Co atoms after the introduction of Ir single atoms. Comparatively, the overpotential at a current density of 10 mA cm⁻² of Ir₁/Cu₀.₃Co₂.₇O₄ with $d = 0.6$ nm were on par with the best records of currently reported Co-based OER catalysts (Supplementary Fig. 18 and Table 2).

To test the generality of distance effect on the stability of cobalt oxide catalysts for acidic OER, we further introduced Ir single atoms

into other spinel cobalt oxides such as Co₃O₄ and Mn₀.₃Co₂.₇O₄. Under similar preparing conditions, we successfully obtained the single-atom catalysts, Ir₁/Co₃O₄ and Ir₁/Mn₀.₃Co₂.₇O₄, with $d = 0.6$ nm (Supplementary Figs. 19–21). The electrochemical measurements revealed that both Ir₁/Co₃O₄ and Ir₁/Mn₀.₃Co₂.₇O₄ also exhibited a high OER stability in acid (Supplementary Fig. 22). This proved that the introduction of Ir single atoms with an appropriate distance into cobalt oxides was a universal strategy to stabilize its structure during acidic OER.

## Discussion

In conclusion, we understand the distance effect of single atoms on the stability of cobalt oxide catalysts for acidic oxygen evolution. Both theoretical calculations and electrocatalytic measurements revealed the stabilizing effect was strongly dependent on the distance of adjacent Ir single atoms. As the distance of adjacent Ir single atoms was reduced to 0.6 nm, the stabilizing effect could cover the Cu₀.₃Co₂.₇O₄, stabilizing entire spinel oxides under acidic conditions. In addition, the introduction of Ir single atoms with an appropriate distance was a universal strategy to stabilize other spinel cobalt oxides during acidic OER. Our work not only provided insight into the distance effect of single atoms on the stability of cobalt oxide catalysts at the atomic level but also pointed toward a direction to the rational design of highly stable catalysts applied in PEMWE.

## Methods
### Chemicals

Cobalt (II) nitrate hexahydrate (Co(NO₃)₂·6H₂O), copper (II) nitrate trihydrate (Cu(NO₃)₂·3H₂O), hexadecyl trimethyl ammonium bromide (CTAB), 2-methylimidazole, active carbon, ethanol (EtOH), perchloric acid (HClO₄), Nafion were purchased from Shanghai Chemical Reagent Company. Iridium (IV) chloride hydrate (IrCl₄·xH₂O) was purchased from Aladdin. All chemicals were of analytical grade and used as received without further purification. All aqueous solutions were prepared using deionized water with a resistivity of 18.2 MΩ cm⁻¹.

## Synthesis of $Cu_{0.3}Co_{2.7}O_4$

$Cu_{0.3}Co_{2.7}O_4$ was synthesized through high-temperature pyrolysis of metal-organic frameworks with modifications[44]. Typically, 497.7 mg of $Co(NO_3)_2\cdot6H_2O$, 70.0 mg of $Cu(NO_3)_2\cdot3H_2O$, and 30.0 mg of CTAB were dissolved in 20 mL of $H_2O$ to form solution A. 9.1 g of 2-methylimidazole was dissolved in 140 mL of $H_2O$ to form solution B. The mixed solution was formed by adding solution A to solution B and mixed for 2 h under magnetic stirring. The resulting solution was centrifuged and washed with EtOH three times to obtain metal-organic frameworks. The product was dried in a vacuum oven overnight. Finally, the above solid was calcined at 350 °C for 4 h in the air to obtain the desired $Cu_{0.3}Co_{2.7}O_4$.

## Synthesis of $Ir_1/Cu_{0.3}Co_{2.7}O_4$ with different Ir single atoms distance

$Ir_1/Cu_{0.3}Co_{2.7}O_4$ with different Ir single atoms distances were synthesized using similar procedures as synthesizing $Cu_{0.3}Co_{2.7}O_4$ except for changing the composition of solution A. For $Ir_1/Cu_{0.3}Co_{2.7}O_4$ with $d = 1.1$ nm, 492.7 mg of $Co(NO_3)_2\cdot6H_2O$, 69.3 mg of $Cu(NO_3)_2\cdot3H_2O$, 6.7 mg of $IrCl_4\cdot xH_2O$, and 30.0 mg of CTAB were dissolved in 20 mL of $H_2O$ to form solution A. For $Ir_1/Cu_{0.3}Co_{2.7}O_4$ with $d = 0.8$ nm, 487.7 mg of $Co(NO_3)_2\cdot6H_2O$, 68.6 mg of $Cu(NO_3)_2\cdot3H_2O$, 13.4 mg of $IrCl_4\cdot xH_2O$, and 30.0 mg of CTAB were dissolved in 20 mL of $H_2O$ to form solution A. For $Ir_1/Cu_{0.3}Co_{2.7}O_4$ with $d = 0.6$ nm, 482.7 mg of $Co(NO_3)_2\cdot6H_2O$, 67.9 mg of $Cu(NO_3)_2\cdot3H_2O$, 20.0 mg of $IrCl_4\cdot xH_2O$, and 30.0 mg of CTAB were dissolved in 20 mL of $H_2O$ to form solution A.

## Synthesis of $Co_3O_4$ and $Mn_{0.3}Co_{2.7}O_4$

$Co_3O_4$ and $Mn_{0.3}Co_{2.7}O_4$ were synthesized using similar procedures as synthesizing $Cu_{0.3}Co_{2.7}O_4$ except for changing the composition of solution A. For $Co_3O_4$, 580.0 mg of $Co(NO_3)_2\cdot6H_2O$ and 30.0 mg of CTAB were dissolved in 20 mL of $H_2O$ to form solution A. For $Mn_{0.3}Co_{2.7}O_4$, 497.7 mg of $Co(NO_3)_2\cdot6H_2O$, 52.9 mg of $Mn(NO_3)_2\cdot6H_2O$, and 30.0 mg of CTAB were dissolved in 20 mL of $H_2O$ to form solution A.

## Synthesis of $Ir_1/Co_3O_4$ and $Ir_1/Mn_{0.3}Co_{2.7}O_4$ with $d = 0.6$ nm

$Ir_1/Co_3O_4$ with $d = 0.6$ nm was synthesized using similar procedures as synthesizing $Co_3O_4$ except for changing the composition of solution A. For $Ir_1/Co_3O_4$ with $d = 0.6$ nm, 482.7 mg of $Co(NO_3)_2\cdot6H_2O$, 20.0 mg of $IrCl_4\cdot xH_2O$, and 30.0 mg of CTAB were dissolved in 20 mL of $H_2O$ to form solution A. $Ir_1/Mn_{0.3}Co_{2.7}O_4$ with $d = 0.6$ nm were synthesized using similar procedures as synthesizing $Mn_{0.3}Co_{2.7}O_4$ except for changing the composition of solution A. For $Ir_1/Mn_{0.3}Co_{2.7}O_4$ with $d = 0.6$ nm, 482.7 mg of $Co(NO_3)_2\cdot6H_2O$, 51.3 mg of $Mn(NO_3)_2\cdot6H_2O$, 20.0 mg of $IrCl_4\cdot xH_2O$, and 30.0 mg of CTAB were dissolved in 20 mL of $H_2O$ to form solution A.

## XAFS measurements

XAFS spectra at Ir $L_3$-edge were obtained at the BL14W1 beamline of Shanghai Synchrotron Radiation Facility (SSRF, Shanghai) operated at 3.5 GeV under 'top-up' mode with a constant current of 220 mA. The XAFS data were recorded under fluorescence mode. The energy was calibrated according to the absorption edge of pure Ir foil. XAFS and in-situ XAFS spectra at Co $K$-edge were obtained at the BL11B beam line of SSRF. The energy was calibrated according to the absorption edge of pure Co foil. Athena software was used to extract the data. For the in-situ XAFS, we performed the experiments in a specialized cell by using a three-electrode standard electrochemical workstation. The catalyst on the carbon substrate was cut into $1.5 \times 1.5$ cm$^2$ pieces and then sealed in a cell by Kapton film. Before the experiments, a series of potentials (OCP ~1.7 V) were applied to the electrode for 2 min, respectively. All XAFS data were collected during one period of beam time and each spectroscopy was recorded for 12 min. XAS spectra at Co $L$-edge, Cu $L$-edge, and O $K$-edge were measured at the beamline BL12B of the National Synchrotron Radiation Laboratory (NSRL, Hefei).

## Electrochemical measurements

An electrochemical workstation (CHI 660E, Shanghai CH Instruments) was used to measure the electrocatalytic performance of the samples. The electrocatalytic measurements were conducted in a standard three-electrode system at room temperature. The carbon paper $(1 \times 0.5$ cm$^{-2})$ loaded with the as-obtained catalysts was used as the working electrode. The mass loadings of catalysts on the carbon paper were 2 mg cm$^{-2}$. A carbon rod was used as the counter electrode. A $Hg/Hg_2SO_4$ electrode was used as the reference electrode. The polarization curves of OER were obtained in 0.1 M $HClO_4$ electrolyte, using a linear sweep voltammetry method at a potential range from 1.02 to 1.82 V with a sweep rate of 5 mV s$^{-1}$. All potentials mentioned in this work were measured against the $Hg/Hg_2SO_4$ electrode and converted to reversible hydrogen electrode (RHE) scale by the equation: $E$ (V vs RHE) = $E$ (V vs $Hg/Hg_2SO_4$) + 0.656 V + 0.0591 pH V. In the given equation, 0.656 V was obtained by calibration with respect to the RHE. The ohmic electrolyte resistance on carbon paper was measured to be 6 Ω. The potentials were corrected to compensate for the effect of solution resistance, which were calculated by the following equation: $E_{iR\text{-corrected}} = E$ (V vs RHE) − $iR$, where $i$ is current, and $R$ is the uncompensated ohmic electrolyte resistance. Tafel slope ($b$) was determined by fitting polarization curves data to the Tafel equation: $\eta = a + b \log |j|$, where $\eta$ is the overpotential for the OER, and $j$ is the current density at the given overpotential. EIS measurements were conducted at 1.55 V. The amplitude of the sinusoidal wave was 5 mV. The frequency scan range was 100 kHz−0.01 Hz. The dissolved Co species were determined by inductively coupled plasma-atomic emission spectrometry (ICP-AES). We first dissolved the pristine $Cu_{0.3}Co_{2.7}O_4$ and $Ir_1/Cu_{0.3}Co_{2.7}O_4$ with $d = 1.1$, 0.8, and 0.6 nm in aqua regia to obtain the mass of the original Co species in catalysts. Then, we conducted electrocatalytic measurements of the pristine $Cu_{0.3}Co_{2.7}O_4$ and $Ir_1/Cu_{0.3}Co_{2.7}O_4$ with different Ir-Ir distances in 100 mL 0.1 M $HClO_4$ electrolyte. Subsequently, we collected 5 mL electrolytes of $Cu_{0.3}Co_{2.7}O_4$ and $Ir_1/Cu_{0.3}Co_{2.7}O_4$ with different Ir-Ir distances at 200, 400, 600, 800, and 1000 scan cycles to obtain the dissolved Co species by conducting ICP-AES test, respectively. Afterward, we added 5 mL of 0.1 M $HClO_4$ to the electrolyte to replenish the volume of the electrolyte. Finally, the dissolution fraction of Co species was obtained by comparing the mass of the remaining Co species to the mass of Co species in pristine catalysts. The experimental procedure for testing dissolved Ir species was similar to that for testing dissolved Co species, except that our experimental method is inductively coupled plasma-mass spectrometry (ICP-MS).

## DFT calculations

Spin-polarized DFT calculations were carried out with Perdew, Burke, and Ernzerhof (PBE) functionals using the Vienna ab initio simulation package (VASP)[45,46]. The projector augmented wave (PAW) method was adopted to describe the ions-electrons interaction[47,48]. A k-point mesh of $3 \times 2 \times 1$ was used to sample the Brillouin zones. The kinetic energy cutoff was set to 400 eV for plane-wave expansion. During structural optimizations, the tolerances of total energy and force were set to $10^{-5}$ eV and 0.05 eV/Å, respectively. To accurately describe the 3$d$ electrons, a correlation energy (U) of 3.0 eV was used for Co and Cu atoms, and 2.0 eV was used for Ir atoms, values that have been tested by previous experimental and theoretical studies[49–51].

## Instrumentations

XRD patterns were recorded using a Philips X'Pert Pro Super diffractometer with Cu-Kα radiation (λ = 1.54178 Å). HAADF-STEM images were taken on a JEOL ARM−200F field-emission transmission electron microscope operating at an accelerating voltage of 200 kV using Mo-

based TEM grids. EDX elemental mapping images were taken on an FEI Talos F200X high-resolution transmission electron microscope using Mo-based TEM grids. ICP-MS (Atomscan Advantage, Thermo Jarrell Ash, USA) analyses were used to determine the mass loadings of Ir single atoms and the dissolved amount of Co species. The distance between Ir single atoms was measured on HAADF-STEM images by Nano Measurer software.

## Data availability

The source data underlying Figs. 1–5 and Supplementary Figs. 1–22 generated in this study are provided as a Source Data file. Source data are provided with this paper.

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

## Acknowledgements

This work was supported by National Key Research and Development Program of China (2021YFA1500500, 2019YFA0405600), CAS Project for Young Scientists in Basic Research (YSBR-051), National Science Fund for Distinguished Young Scholars (21925204), NSFC (22202192, U19A2015, 22221003, 22250007, 22163002), Collaborative Innovation Program of Hefei Science Center, CAS (2022HSC-CIP004), International Partnership Program of Chinese Academy of Sciences (123GJHZ2022101GC), the DNL Cooperation Fund, CAS (DNL202003), Anhui Natural Science Foundation for Young Scholars (2208085QB52), USTC Research Funds of the Double First-Class Initiative (YD9990002016), the Guizhou Provincial Science and Technology Projects (QKHJC-ZK[2021]YB047, 2021GZJ001), Fellowship of China Postdoctoral Science Foundation (2023M743372), and Postdoctoral Fellowship Program of CPSF (GZC20232537).

## Author contributions

Z.Z., S.Z., and J.Z. designed the study. P.M., C.F., J.Y., J.H., J.Zheng, M.L., and Z.Z. conducted the experiments. M.Z. conducted HAADF-STEM analysis. C.J. carried out DFT calculations. Z.Z., S.Z., and J.Z. wrote the paper. All authors discussed the results and contributed to the manuscript.

## Competing interests

The authors declare no competing interests.
