## [Peer Review File · Nature Communications]

Distance effect of single atoms on stability of cobalt oxide catalysts for acidic oxygen evolutionREVIEWER COMMENTS

Reviewer #1 (Remarks to the Author):

In this paper, Zhang et al. addresses one of the challenges of the development of PEM water electrolyzers (PEMWEs), which is the scarcity of Ir resources. Specifically, the work reports on the typical instability of Ir-free PEMWE catalysts, $\text{Cu}_x\text{Co}_y\text{O}_z$, and how the insertion of Ir into this type of category of catalysts stabilizes them. Zhang et al. prepared several $\text{Cu}_x\text{Co}_y\text{O}_z$ catalysts with specific Ir contents, where they controlled the Ir-Ir distance. They demonstrated that the decrease in the Ir-Ir distance results in higher stability of the catalyst. The scope of the work is definitely of significance for catalyst development in the field of PEMWE, however, there are major concerns in the manuscript (especially in the electrochemical characterization section) that makes it challenging for this manuscript to be published in Nature Communications.

1. The authors properly addressed general issues regarding Ir resources and how it affects PEMWE technology and cited suitable references in the introduction section. However, it is advised to add a paragraph in this section regarding the general strategies to decrease Ir loading in PEMWE, such as deposition of IrO_2 on support materials like TiO_2 , and so on.
2. Although the DFT and surface characterization studies were very comprehensive, the electrochemical investigations missed considering some of the main pillars of OER measurements. These include the use of carbon at high potential where a significant corrosion rate takes place depending on the applied potential (see point 9). Also, the authors reported the use of a stagnant electrode for a gas evolving reaction, which results in accumulation of the gas bubbles resulting in shielding of the active sites and ending up with unreliable results. In general, a rotating disk electrode or a flow-through technique should be used for such studies in order to efficiently remove the oxygen bubbles.
3. Fixing the catalyst loading at $2\text{mg}/\text{cm}^2$ is not the best approach to use in this study as it confuses the effect of Ir-Ir distance. By decreasing the Ir-Ir distance in the catalyst, and still using the same loading on the support, this means that the Ir-Ir distance decrease is associated with an increase of the Ir content on the electrode. This in turn results in higher activity and makes the comparison between different electrode not valid anymore as we are not looking now into only the effect of Ir-Ir distance, but also at the Ir loading which differs from one catalyst to the next.
4. As the main focus of this manuscript is the stability of the catalyst as a function of the Ir-Ir distance, it is important to mention that Ir is the second most active metal towards OER, therefore, the smaller the Ir-Ir distance, while still using the same loading of the catalyst on the support, the higher the Ir content on the electrode. This in turn results in most of the OER reaction takes place on the Ir and therefore

there is less stress on the other components and therefore they show less dissolution. The same thing applies in a chronoamperometry test, a catalyst with the smallest Ir-Ir distance, will have the highest Ir loading (in the 2 mg/cm²), and therefore most of the OER reaction takes place on the Ir resulting in less dissolution of the other components. In short, I believe there is an artifact in the measurements which makes the results questionable.

5. Generally, it is not clear why Cu_{0.3}Co_{2.7}O₄ is selected as a model system and most of the manuscript is created around this starting material, specially that Cu is also an unstable element under acidic conditions. Why didn't the authors simply choose Co₃O₄ as the model system and continue with it?

6. There is no theoretical calculation to base the synthesis recipe in the manuscript. How did the authors come up with the composition of the precursors to end up with specific Ir-Ir distances?

7. The term "support" is occasionally used to describe Cu_{0.3}Co_{2.7}O₄ in the manuscript (e.g. line 166 in the manuscript). It is suggested to replace "support" with another term to prevent confusion.

8. It is recommended to show the original TEM images of the samples in Fig. S5.

9. The authors have used a carbon paper as the substrate for the electrochemical measurements. Since carbon corrosion is severe at the conditions that the electrochemical measurements have been performed, specially during the durability experiments (pH=1, E > 1.5 V vs RHE), how do the authors exclude the effect of substrate carbon paper corrosion and the detachment of catalyst? This problem becomes more severe when the tested catalysts have significant differences in the OER activity. For example in Fig. 5d, at a constant current of 10 mA.cm⁻², the red and blue curves have a difference of ≈100 mV at the beginning of test. This means that the rate of carbon corrosion is significantly higher for the blue curve compared to the red curve, which could result in faster degradation of the performance due to the catalyst detachment.

10. How was the Co dissolution measured in Fig. 5c. Please provide details about sample solution collection during the experiments.

11. In Fig. 5b, the decrease in j at which potential is presented?

12. What was the cell setup for the in-situ XAFS measurements? Did the authors use the same cell as they used for electrochemical measurements? Please provide further detail regarding the experimental

procedure in this section, such as applied time for each potential and so on. Why didn't authors test the reference $\text{Cu}_0.3\text{Co}_2.7\text{O}_4$ material under the same condition to prove the stability improvement by Ir incorporation?

13. The potential values are reported vs. RHE. Is this based on a theoretical calculation or an actual calibration of $\text{Hg}/\text{Hg}_2\text{SO}_4$ vs. a hydrogen electrode?

Reviewer #2 (Remarks to the Author):

In this work, Zhang et al. synthesized a series of spinel oxide catalysts with different distance atomic iridium species anchored. They proved that the Ir single atoms had a stabilizing effect on the spinel oxides. More importantly, they proposed that the stabilizing effect was strongly related to the distance between Ir single atoms, for the first time. The closer the distance between the single atoms, the stronger the stabilizing effect on the catalysts. When the distance between Ir single atoms reached 0.6 nm, the stabilizing effect covered the $\text{Cu}_0.3\text{Co}_2.7\text{O}_4$, stabilizing entire spinel oxides under acidic conditions. The work is noteworthy and proposes a new insight into the rational design of highly stable catalysts applied in PEMWE. I believe this manuscript is very worthy of publication in Nature Communications after minor revisions.

1. Differences in the anchoring sites of single atoms may lead to diversities in the intensities of the stabilizing effect induced by single atoms on spinel oxides. It is necessary for the authors to point out the anchoring site of the Ir single atoms in $\text{Ir}_1/\text{Cu}_0.3\text{Co}_2.7\text{O}_4$.

2. The authors experimentally verified that Ir single atoms have a stabilizing effect on $\text{Cu}_0.3\text{Co}_2.7\text{O}_4$ during acidic OER and proved that the stabilizing effect was related to the distance between Ir single atoms. However, the source of the stabilizing effect was barely discussed in the manuscript. It is recommended to discuss the origin of this stabilizing effect in detail.

3. In Supplementary Fig. 4, after introducing Ir single atoms into the Co_3O_4 , the characteristic peaks in the Raman spectra of $\text{Ir}_1/\text{Cu}_0.3\text{Co}_2.7\text{O}_4$ with different Ir-Ir distances were all shifted. It is recommended that the authors explain this shift in the manuscript.

4. The authors claimed the obtained $\text{Ir}_1/\text{Cu}_0.3\text{Co}_2.7\text{O}_4$ with $d = 0.6$ nm exhibits excellent acidic OER catalytic stability. Catalyst stability should be compared to commercial IrO_2 stability toward acidic OER.

5. After the durability test, the lattice fringe of the $\text{Ir}_1/\text{Cu}_0.3\text{Co}_2.7\text{O}_4$ with $d = 0.6$ nm became ambiguous. Please explain the origin of the phenomenon.

6. After introducing the Ir single atoms with $d = 0.6$ nm to the $\text{Cu}_0.3\text{Co}_2.7\text{O}_4$, the $\text{Ir}_1/\text{Cu}_0.3\text{Co}_2.7\text{O}_4$ presented no obvious degradation over a 60-hour stability test for acidic OER. To further illustrate the excellent stability of this sample, please add the change in the applied voltage after stabilization to Fig. 5d.

7. Did the catalytic performance of Ir₁/Cu_{0.3}Co_{2.7}O₄ with d = 0.6 nm toward acidic OER lead among the spinel oxides? It would be better for the authors to give a figure to compare the catalytic performance of Ir₁/Cu_{0.3}Co_{2.7}O₄ with d = 0.6 nm with some other spinel oxides.

Reviewer #3 (Remarks to the Author):

Point-by-point response to reviewers' comments

Manuscript Number: NCOMMS-23-43865

Manuscript Type: Research Article

Title: "Distance effect of single atoms on the stability of cobalt oxide catalysts for acidic oxygen evolution"

Reviewer #1 (Remarks to the Author):

In this paper, Zhang et al. addresses one of the challenges of the development of PEM water electrolyzers (PEMWEs), which is the scarcity of Ir resources. Specifically, the work reports on the typical instability of Ir-free PEMWE catalysts, $\text{Cu}_x\text{Co}_y\text{O}_z$, and how the insertion of Ir into this type of category of catalysts stabilizes them. Zhang et al. prepared several $\text{Cu}_x\text{Co}_y\text{O}_z$ catalysts with specific Ir contents, where they controlled the Ir-Ir distance. They demonstrated that the decrease in the Ir-Ir distance results in higher stability of the catalyst. The scope of the work is definitely of significance for catalyst development in the field of PEMWE, however, there are major concerns in the manuscript (especially in the electrochemical characterization section) that makes it challenging for this manuscript to be published in Nature Communications.

Response: Thanks to the reviewers for giving valuable advice. Taking into account the ill-considered methods in the original electrocatalytic evaluation process, we have optimized the process of electrochemical evaluation. Specifically, we replaced the carbon paper with Ti felt and removed activated carbon in the electrode preparation process to avoid the influence of disturbing factors on the results.

1. The authors properly addressed general issues regarding Ir resources and how it affects PEMWE technology and cited suitable references in the introduction section. However, it is advised to add a paragraph in this section regarding the general strategies to decrease Ir loading in PEMWE, such as deposition of IrO_2 on support materials like TiO_2 , and so on.

Response: Thank this reviewer for the valuable advice to improve the quality of this work. We have added a paragraph into the introduction section of our manuscript to discuss the general strategies to decrease Ir loading in PEMWE.

The content is as follows: Great efforts have been devoted to exploring effective strategies to decrease iridium loadings in acidic OER catalysts, such as constructing Ir-metal clusters, alloying, and deposition of iridium species on suitable support. However, these strategies still utilize relatively abundant iridium species. Therefore, it is highly desired but remains a major challenge, to develop efficient and durable catalysts with low Ir consumption for acidic OER.

2. Although the DFT and surface characterization studies were very comprehensive, the electrochemical investigations missed considering some of the main pillars of OER measurements. These include the use of carbon at high potential where a significant corrosion rate takes place depending on the applied potential (see point 9). Also, the authors reported the use of a stagnant electrode for a gas evolving reaction, which results in accumulation of the gas

bubbles resulting in shielding of the active sites and ending up with unreliable results. In general, a rotating disk electrode or a flow-through technique should be used for such studies in order to efficiently remove the oxygen bubbles.

Response: We genuinely thank the reviewer for the valuable concern. We agree with the reviewers' comments that the carbon corrosion may influence the electrochemical test results. To exclude the influence of carbon corrosion on the acidic OER measurements, we replaced the catalyst support from carbon paper with Ti felt and active carbon was excluded during the electrode preparation stage. As shown in Fig. R1a, the current density of pristine $\text{Cu}_{0.3}\text{Co}_{2.7}\text{O}_4$ showed an obvious decrease as the number of scans increased. As Ir single atoms with $d = 1.1$ and 0.8 nm were introduced into $\text{Cu}_{0.3}\text{Co}_{2.7}\text{O}_4$, the decline rate in the current density decrease slowed down, indicating the Ir single atoms stabilized the support (Fig. R1, b, c, and R2). When the Ir-Ir distance was further reduced to 0.6 nm, the current density showed an unspectacular decrease during the sequential scans, indicating excellent OER stability for $\text{Ir}_1/\text{Cu}_{0.3}\text{Co}_{2.7}\text{O}_4$ with $d = 0.6$ nm in acidic media (Fig. R1d and R2). The results indicated that the stabilizing effect induced by Ir single atoms was dependent on the distance of adjacent Ir single atoms whether choosing carbon paper or Ti felt as the support.

To avoid gas bubbles shielding the active sites, we also evaluated the OER activity of $\text{Cu}_{0.3}\text{Co}_{2.7}\text{O}_4$ and $\text{Ir}_1/\text{Cu}_{0.3}\text{Co}_{2.7}\text{O}_4$ with $d = 0.6$ nm in a flow-cell setup with 0.1 M HClO_4 electrolyte (Fig. R3a). According to the polarization curves, as the number of scans increases, the current density of pristine $\text{Cu}_{0.3}\text{Co}_{2.7}\text{O}_4$ decreases significantly (Fig. R3b). When Ir single atoms with $d = 0.6$ nm were introduced into the $\text{Cu}_{0.3}\text{Co}_{2.7}\text{O}_4$, the current density of the $\text{Ir}_1/\text{Cu}_{0.3}\text{Co}_{2.7}\text{O}_4$ with $d = 0.6$ nm showed an inconspicuous decrease during the scans, indicating excellent stability for this sample in acidic media (Fig. R3c). The above results demonstrated that the Ir single atoms with suitable distance can stabilize the entire $\text{Cu}_{0.3}\text{Co}_{2.7}\text{O}_4$.

The correlated discussions have been added to our manuscript.

Fig. R1 | Stability evaluation towards acidic oxygen evolution on Ti felt. a-d, Polarization curves of $\text{Cu}_{0.3}\text{Co}_{2.7}\text{O}_4$ (a) and $\text{Ir}_1/\text{Cu}_{0.3}\text{Co}_{2.7}\text{O}_4$ with $d = 1.1$ nm (b), $d = 0.8$ nm (c), and $d = 0.6$ nm (d) at different scan cycles in 0.1 M HClO_4 electrolyte. The displayed polarization curves are the 1, 200, 400, 600, 800, and 1000 cycles, respectively. No activated carbon was added at the electrode preparation stage.

Fig. R2 | Decrease in current densities under different scan cycles of $\text{Cu}_{0.3}\text{Co}_{2.7}\text{O}_4$ and $\text{Ir}_1/\text{Cu}_{0.3}\text{Co}_{2.7}\text{O}_4$ with different Ir-Ir distances on Ti felt. The selected overpotential was 1.70 V (E vs RHE) for all catalysts.

Fig. R3 | Stability evaluation of Ir₁/Cu_{0.3}Co_{2.7}O₄ with d = 0.6 nm for acidic OER in a flow-cell setup. a, Optical image of the flow-cell setup. **b, c,** Polarization curves of Cu_{0.3}Co_{2.7}O₄ (**b**) and Ir₁/Cu_{0.3}Co_{2.7}O₄ with d = 0.6 nm (**c**) at different scan cycles in 0.1 M HClO₄ electrolyte. The displayed polarization curves are the 1, 200, 400, 600, 800, and 1000 cycles, respectively.

3. Fixing the catalyst loading at 2mg/cm² is not the best approach to use in this study as it confuses the effect of Ir-Ir distance. By decreasing the Ir-Ir distance in the catalyst, and still using the same loading on the support, this means that the Ir-Ir distance decrease is associated with an increase of the Ir content on the electrode. This in turn results in higher activity and makes the comparison between different electrode not valid anymore as we are not looking now into only the effect of Ir-Ir distance, but also at the Ir loading which differs from one catalyst to the next.

Response: We honestly thank the reviewer for raising this important issue. We agree with the reviewers' comments that the different Ir content on the electrode may influence the electrochemical performance. To study this influence, we carried out the electrochemical measurements on the catalysts by fixing the Ir loadings on the electrode at 25 μg cm⁻², as shown in Fig. R4. The polarization curves at the 1st cycle revealed that Ir₁/Cu_{0.3}Co_{2.7}O₄ with d = 1.1, 0.8, and 0.6 nm required an overpotential of 350, 340, and 360 mV to deliver a current density of 10 mA cm⁻², respectively. These values were indeed different from those on the electrode with fixing the catalyst loadings at 2 mg cm⁻². However, the stability evaluation showed that the current density after 1000 scan cycles significantly decreased for Ir₁/Cu_{0.3}Co_{2.7}O₄ with d = 1.1 and 0.8 nm, but almost remained unchanging for Ir₁/Cu_{0.3}Co_{2.7}O₄ with d = 0.6 nm. These results are very similar to those on the electrode with fixing the catalyst loadings at 2 mg cm⁻². This feature indicated that although the OER activities of these catalysts were influenced by the Ir content on the electrode, the OER stability in acid was unaffected by the Ir content on the electrode. All these results further proved that the stabilizing effect induced by Ir single atoms was only related to the distance of adjacent Ir single atoms, but not to the Ir content on the electrode.

The correlated discussions have been added to our manuscript.

Fig. R4 | Stability evaluation towards acidic oxygen evolution of catalysts on Ti felt. a-c, Polarization curves of Ir₁/Cu_{0.3}Co_{2.7}O₄ with d = 1.1 nm (a), d = 0.8 nm (b), and d = 0.6 nm (c) at different scan cycles in 0.1 M HClO₄ electrolyte. The loadings of Ir species on the electrode were fixed at 25 $\mu\text{g cm}^{-2}$. The displayed polarization curves are the 1, 200, 400, 600, 800, and 1000 cycles, respectively. No activated carbon was added at the catalyst preparation stage. d, Decrease in current densities under different scan cycles of Ir₁/Cu_{0.3}Co_{2.7}O₄ with d = 1.1 nm, 0.8 nm, and 0.6 nm containing similar loadings of Ir species. The selected overpotential was 1.70 V (*E* vs RHE) for all catalysts.

4. As the main focus of this manuscript is the stability of the catalyst as a function of the Ir-Ir distance, it is important to mention that Ir is the second most active metal towards OER, therefore, the smaller the Ir-Ir distance, while still using the same loading of the catalyst on the support, the higher the Ir content on the electrode. This in turn results in most of the OER reaction takes place on the Ir and therefore there is less stress on the other components and therefore they show less dissolution. The same thing applies in a chronoamperometry test, a catalyst with the smallest Ir-Ir distance, will have the highest Ir loading (in the 2 mg/cm^2), and therefore most of the OER reaction takes place on the Ir resulting in less dissolution of the other components. In short, I believe there is an artifact in the measurements which makes the results questionable.

Response: We genuinely thank the reviewer for the valuable comment. As discussed above, our additional electrochemical measurements (Fig. R4) on the catalysts by fixing the Ir loadings on the electrode at 25 $\mu\text{g cm}^{-2}$ showed that although the OER activities of these catalysts were influenced by the Ir content on the electrode, the OER stability in acid was unaffected by the Ir content on the electrode. This suggested that the stabilizing effect induced by Ir single atoms was strongly dependent on the distance of adjacent Ir single atoms, while not related to the content of Ir single atoms on the electrode.

5. Generally, it is not clear why $\text{Cu}_{0.3}\text{Co}_{2.7}\text{O}_4$ is selected as a model system and most of the manuscript is created around this starting material, specially that Cu is also an unstable element under acidic conditions. Why didn't the authors simply choose Co_3O_4 as the model system and continue with it?

Response: We sincerely thank this reviewer for the insightful questions. We first synthesized Co_3O_4 and $\text{Cu}_{0.3}\text{Co}_{2.7}\text{O}_4$ and evaluated their catalytic performance in acidic oxygen evolution. As shown in the polarization curves, the $\text{Cu}_{0.3}\text{Co}_{2.7}\text{O}_4$ showed higher OER activities than Co_3O_4 (Fig. R5a). The former required an overpotential of 410 mV to deliver current densities of 10 mA cm^{-2} , which were 40 mV lower than that of the latter. Furthermore, we also evaluated the charge-transfer resistances of Co_3O_4 and $\text{Cu}_{0.3}\text{Co}_{2.7}\text{O}_4$ by electrochemical impedance spectroscopy (EIS) since the charge transfer rate at the catalyst interface plays a crucial role in catalyst performance (Fig. R5b). The semicircle diameter of $\text{Cu}_{0.3}\text{Co}_{2.7}\text{O}_4$ was smaller than that of Co_3O_4 , indicating its faster charge transfer, which favors a more rapid reaction rate. Based on the catalytic performance, we selected $\text{Cu}_{0.3}\text{Co}_{2.7}\text{O}_4$ as the model system in this work.

Fig. R5 | Electrocatalytic measurements. **a**, Polarization curves of Co_3O_4 and $\text{Cu}_{0.3}\text{Co}_{2.7}\text{O}_4$ in 0.1 M HClO_4 electrolyte. **b**, Electrochemical impedance spectra of Co_3O_4 and $\text{Cu}_{0.3}\text{Co}_{2.7}\text{O}_4$.

6. There is no theoretical calculation to base the synthesis recipe in the manuscript. How did the authors come up with the composition of the precursors to end up with specific Ir-Ir distances?

Response: We genuinely thank this reviewer for the insightful questions. At first, we synthesized a series of $\text{Ir}_1/\text{Cu}_{0.3}\text{Co}_{2.7}\text{O}_4$ catalysts with different Ir-Ir distances of 1.1, 1.0 (Fig. R6a), 0.8, 0.7 (Fig. R6b), and 0.6 nm. Considering the Ir-Ir distances of the optimized structural models $\text{Ir}_1/\text{Cu}_{0.3}\text{Co}_{2.7}\text{O}_4$ were 1.14, 0.87, and 0.56 nm, we selected $\text{Ir}_1/\text{Cu}_{0.3}\text{Co}_{2.7}\text{O}_4$ with $d = 1.1, 0.8,$ and 0.6 nm as the research targets, which had the similar Ir-Ir distances with optimized structural models. In addition, the choice of the research targets also considered the length restriction of the manuscript.

Fig. R6 | Morphology characterization. a, b, HAADF-STEM images of $\text{Ir}_1/\text{Cu}_{0.3}\text{Co}_{2.7}\text{O}_4$ with $d = 1.0$ (a) and 0.7 nm (b).

7. The term "support" is occasionally used to describe $\text{Cu}_{0.3}\text{Co}_{2.7}\text{O}_4$ in the manuscript (e.g. line 166 in the manuscript). It is suggested to replace "support" with another term to prevent confusion.

Response: We honestly thank the reviewer for the valuable suggestion. We have replaced "support" with $\text{Cu}_{0.3}\text{Co}_{2.7}\text{O}_4$ in our manuscript. Thank you.

8. It is recommended to show the original TEM images of the samples in Fig. S5.

Response: Thank this reviewer for the valuable advice to improve the quality of this work. We have added the original TEM images of the samples to Supplementary Fig. S5. The TEM images showed that the as-prepared $\text{Ir}_1/\text{Cu}_{0.3}\text{Co}_{2.7}\text{O}_4$ with different Ir-Ir distances presented similar morphologies (Fig. R7, a, c, e). Energy dispersive X-ray (EDX) elemental mapping images revealed that the Ir atoms were uniformly distributed in all three $\text{Ir}_1/\text{Cu}_{0.3}\text{Co}_{2.7}\text{O}_4$ samples (Fig. R7, b, d, f).

Fig. R7 | Morphology characterization. **a, c, e**, TEM images of Ir₁/Cu_{0.3}Co_{2.7}O₄ with d = 1.1 nm (**a**), 0.8 nm (**c**), and 0.6 nm (**e**). **b, d, f**, EDX elemental mapping of Ir₁/Cu_{0.3}Co_{2.7}O₄ with d = 1.1 nm (**b**), 0.8 nm (**d**), and 0.6 nm (**f**).

9. The authors have used a carbon paper as the substrate for the electrochemical measurements. Since carbon corrosion is severe at the conditions that the electrochemical measurements have been performed, specially during the durability experiments (pH=1, E > 1.5 V vs RHE), how do the authors exclude the effect of substrate carbon paper corrosion and the detachment of catalyst? This problem becomes more severe when the tested catalysts have significant differences in the OER activity. For example in Fig. 5d, at a constant current of 10 mA cm⁻², the red and blue curves have a difference of ≈100 mV at the beginning of test. This means that the rate of carbon corrosion is significantly higher for the blue curve compared to the red curve, which could result in faster degradation of the performance due to the catalyst detachment.

Response: We genuinely thank the reviewer for the insightful comment. We agree with the reviewers' comments that carbon corrosion may influence the electrochemical measurements. To avoid the effects of substrate carbon paper corrosion on the experiment results, we evaluated the catalytic performance and stability of these catalysts by replacing the carbon paper with Ti felt. In addition, activated carbon was also not added during the electrode preparation stage. As shown in Fig. R1a, the current density of pristine Cu_{0.3}Co_{2.7}O₄ showed an obvious decrease after 1000 scan cycles, demonstrating its poor stability in acidic media. As Ir single atoms with d = 1.1 and 0.8 nm were introduced into Cu_{0.3}Co_{2.7}O₄, the decline rate in the current density decrease slowed down, indicating the Ir single atoms could stabilize the support (Fig. R1, b and c). When the Ir-Ir distance was further reduced to 0.6 nm, the current density showed an inconspicuous decrease during the sequential scans, indicating excellent OER stability for this sample in acidic media (Fig. R1d). The above results were consistent with the results of experiments conducted on carbon paper, indicating the stabilizing effect induced by Ir single atoms was dependent on the distance of adjacent Ir single atoms.

10. How was the Co dissolution measured in Fig. 5c. Please provide details about sample solution collection during the experiments.

Response: We genuinely thank this reviewer for the insightful questions. The dissolved Co species were determined by inductively coupled plasma-atomic emission spectrometry (ICP-AES). We first dissolved the pristine Cu_{0.3}Co_{2.7}O₄ and Ir₁/Cu_{0.3}Co_{2.7}O₄ with d = 1.1, 0.8, and 0.6 nm in aqua regia to obtain the mass of the original Co species in catalysts. Then, we conducted electrocatalytic measurements of the pristine Cu_{0.3}Co_{2.7}O₄ and Ir₁/Cu_{0.3}Co_{2.7}O₄ with different Ir-Ir distances in 100 mL 0.1 M HClO₄ electrolyte. Subsequently, we collected 5 mL electrolytes of Cu_{0.3}Co_{2.7}O₄ and Ir₁/Cu_{0.3}Co_{2.7}O₄ with different Ir-Ir distances at 200, 400, 600, 800, and 1000 scan cycles to obtain the dissolved Co species by conducting ICP-AES test, respectively. Afterward, we added 5 mL of 0.1 M HClO₄ to the electrolyte to replenish the volume of the electrolyte. Finally, the dissolution fraction of Co species was obtained by comparing the mass of the remaining Co species to the mass of Co species in pristine catalysts.

The experiment details have been added to the electrochemical measurements section in our manuscript.

11. In Fig. 5b, the decrease in j at which potential is presented?

Response: We truly thank the reviewer for raising this important issue. In the original Fig. 5b, we chose the voltage at which the current density reaches 30 mA cm^{-2} in the first cycle as the reference to evaluate the magnitude of the current density decrease. To further assess the stability of these catalysts, we also compared the current densities at a constant potential of 1.60 V during cycling for these catalysts. As shown Fig. R8, the current density of pristine $\text{Cu}_{0.3}\text{Co}_{2.7}\text{O}_4$ decreased by 87.5% after 1000 scan cycles, and they decreased by 76.3% and 44.2% for $\text{Ir}_1/\text{Cu}_{0.3}\text{Co}_{2.7}\text{O}_4$ with $d = 1.1$ and 0.8 nm, respectively. When the Ir-Ir distance was reduced to 0.6 nm, the current density showed an inconspicuous decrease during the sequential scans. These results are very similar to those shown in the original Fig. 5b, suggesting that the OER stability depended on the Ir-Ir distances.

To harmonize the criteria of stability assessment, we have added the relevant data and correlated discussions to our manuscript as Fig. 5b.

Fig. R8 | Decrease in current densities under different scan cycles of $\text{Cu}_{0.3}\text{Co}_{2.7}\text{O}_4$ and $\text{Ir}_1/\text{Cu}_{0.3}\text{Co}_{2.7}\text{O}_4$ with different Ir-Ir distances. The selected overpotential was 1.60 V (E vs RHE) for all catalysts.

12. What was the cell setup for the in-situ XAFS measurements? Did the authors use the same cell as they used for electrochemical measurements? Please provide further detail regarding the experimental procedure in this section, such as applied time for each potential and so on. Why didn't authors test the reference $\text{Cu}_{0.3}\text{Co}_{2.7}\text{O}_4$ material under the same condition to prove the stability improvement by Ir incorporation?

Response: We sincerely thank this reviewer for the insightful questions. For the electrochemical measurements, we carried out the measurements in a standard three-electrode system at an electrochemical cell (Fig. R9).

For the *in-situ* XAFS, we performed the experiments in a specialized cell by using a three-electrode standard electrochemical workstation (Fig. R10a). The XAFS spectra were collected through fluorescence mode (Fig. R10b). The catalyst on the carbon substrate was cut into $1.5 \times 1.5 \text{ cm}^2$ pieces and then sealed in a cell by Kapton film. Before the experiments, a series of potentials (open circle potential $\sim 1.7 \text{ V}$) were applied to the electrode for 2 min, respectively. All XAFS data were collected during one period of beam time and each spectroscopy was recorded for 12 min.

We also performed *in-situ* XAFS to evaluate the stability of $\text{Cu}_{0.3}\text{Co}_{2.7}\text{O}_4$ during acidic OER. As shown in Fig. R10c, the absorption edge of Co *K*-edge XANES spectra for $\text{Cu}_{0.3}\text{Co}_{2.7}\text{O}_4$ exhibited inconspicuous changes with increasing applied voltage from OCP to 1.7 V. However, the EXAFS spectra revealed that the intensity of the two main peaks from the Co-O and $\text{Co}_{\text{oct}}\text{-Co}_{\text{oct}}$ coordination exhibited a decreasing tendency with increasing applied voltage, indicating the $\text{Cu}_{0.3}\text{Co}_{2.7}\text{O}_4$ was gradually dissolved during acidic OER (Fig. R10d). For comparison, *in-situ* XAFS at the Co *K*-edge of $\text{Ir}_1/\text{Cu}_{0.3}\text{Co}_{2.7}\text{O}_4$ with $d = 0.6 \text{ nm}$ was also conducted. The absorption edge of Co *K*-edge XANES spectra and the main peaks from the Co-O and $\text{Co}_{\text{oct}}\text{-Co}_{\text{oct}}$ coordination in EXAFS spectra for $\text{Ir}_1/\text{Cu}_{0.3}\text{Co}_{2.7}\text{O}_4$ with $d = 0.6 \text{ nm}$ both exhibited negligible changes with increasing voltage from OCP to 1.7 V, suggesting the excellent stability of the catalysts under oxidative potentials (Fig. R10, e and f). The experiment results revealed that Ir single atoms stabilized $\text{Cu}_{0.3}\text{Co}_{2.7}\text{O}_4$ under acidic OER conditions. The correlated discussions have been added to our manuscript.

Fig. R9 | Electrochemical characterizations. Optical image of the standard three-electrode system at an electrochemical cell.

Fig. R10 | *In-situ* spectroscopic characterizations. a, b, Optical image of the *in-situ* XAFS electrolytic cell (a) and the device for *in-situ* XAFS measurement (b). c, d, *In-situ* Co *K*-edge XANES (c) and EXAFS (d) spectra of $\text{Cu}_{0.3}\text{Co}_{2.7}\text{O}_4$ at different applied potentials. e, f, *In-situ* Co *K*-edge XANES (e) and EXAFS (f) spectra of $\text{Ir}_1/\text{Cu}_{0.3}\text{Co}_{2.7}\text{O}_4$ with $d = 0.6$ nm at different applied potentials.

13. The potential values are reported vs. RHE. Is this based on a theoretical calculation or an actual calibration of $\text{Hg}/\text{Hg}_2\text{SO}_4$ vs. a hydrogen electrode?

Response: We genuinely thank the reviewer for the valuable comment. All potentials mentioned in this work were measured against the $\text{Hg}/\text{Hg}_2\text{SO}_4$ electrode and converted to reversible hydrogen electrode (RHE) scale by the equation: E (V vs RHE) = E (V vs $\text{Hg}/\text{Hg}_2\text{SO}_4$) + 0.656 V + 0.0591pH V. In the given equation, 0.656V was obtained by calibration with respect to the RHE. The calibration was carried out in a three-electrode system using hydrogen-saturated 0.1 M HClO_4 as the electrolyte, where a platinum wire, another platinum wire, and an $\text{Hg}/\text{Hg}_2\text{SO}_4$ electrode were used as the working, counter, and reference electrodes, respectively. The calibration was conducted using a cyclic voltammetry method with a sweep rate of 1 mV s^{-1} . The average value of the intercept on the potential axis was considered the thermodynamic equilibrium potential for the hydrogen electrode reactions. The experiment details have been added to the electrochemical measurements section in our manuscript.

Reviewer #2 (Remarks to the Author):

In this work, Zhang et al. synthesized a series of spinel oxide catalysts with different distance atomic iridium species anchored. They proved that the Ir single atoms had a stabilizing effect on the spinel oxides. More importantly, they proposed that the stabilizing effect was strongly related to the distance between Ir single atoms, for the first time. The closer the distance between the

single atoms, the stronger the stabilizing effect on the catalysts. When the distance between Ir single atoms reached 0.6 nm, the stabilizing effect covered the $\text{Cu}_{0.3}\text{Co}_{2.7}\text{O}_4$, stabilizing entire spinel oxides under acidic conditions. The work is noteworthy and proposes a new insight into the rational design of highly stable catalysts applied in PEMWE. I believe this manuscript is very worthy of publication in Nature Communications after minor revisions.

1. Differences in the anchoring sites of single atoms may lead to diversities in the intensities of the stabilizing effect induced by single atoms on spinel oxides. It is necessary for the authors to point out the anchoring site of the Ir single atoms in $\text{Ir}_1/\text{Cu}_{0.3}\text{Co}_{2.7}\text{O}_4$.

Response: We sincerely thank the reviewer for the valuable comment. In Fig. 4d, the EXAFS spectra of $\text{Ir}_1/\text{Cu}_{0.3}\text{Co}_{2.7}\text{O}_4$ with $d = 1.1, 0.8,$ and 0.6 nm discerned a characteristic peak at approximately 3.1 \AA , which was ascribed to the Ir-Co bonding from the second-shell coordination. This result confirmed that the Ir single atoms of $\text{Ir}_1/\text{Cu}_{0.3}\text{Co}_{2.7}\text{O}_4$ with different Ir-Ir distances were incorporated into the lattice of $\text{Cu}_{0.3}\text{Co}_{2.7}\text{O}_4$.

2. The authors experimentally verified that Ir single atoms have a stabilizing effect on $\text{Cu}_{0.3}\text{Co}_{2.7}\text{O}_4$ during acidic OER and proved that the stabilizing effect was related to the distance between Ir single atoms. However, the source of the stabilizing effect was barely discussed in the manuscript. It is recommended to discuss the origin of this stabilizing effect in detail.

Response: We honestly thank this reviewer for the insightful questions. Generally, IrO_2 is considered to be the most stable electrocatalyst for the acidic OER. The source of the excellent stability for IrO_2 originated from the strong metal-oxygen bonding strength (*Nat. Commun.* **14**, 5365 (2023); *Nat. Commun.* **14**, 354 (2023)). When the Ir single atoms were introduced into the $\text{Cu}_{0.3}\text{Co}_{2.7}\text{O}_4$, the Ir-O-Co local structure could be constructed, thus improving the stability of Co sites in electrocatalysts. The Ir L_3 -edge EXAFS spectra exhibited two characteristic peaks at about 2.0 \AA and 3.0 \AA , which were ascribed to first-shell Ir-O coordination and second-shell Ir-Co coordination, respectively, confirming the presence of the Ir-O-Co structure (Fig. 4d). As the distance of adjacent Ir single atoms narrowed to a suitable value, the Ir-O-Co structure covered the entire $\text{Cu}_{0.3}\text{Co}_{2.7}\text{O}_4$, making the stabilizing effect superimposed on each other, thus significantly stabilizing the entire spinel oxides. The relevant discussion has been added to our manuscript.

3. In Supplementary Fig. 4, after introducing Ir single atoms into the Co_3O_4 , the characteristic peaks in the Raman spectra of $\text{Ir}_1/\text{Cu}_{0.3}\text{Co}_{2.7}\text{O}_4$ with different Ir-Ir distances were all shifted. It is recommended that the authors explain this shift in the manuscript.

Response: We genuinely thank the reviewer for the insightful comment. Generally, the shift of characteristic peaks in the Raman spectra was closely related to the modifications of the catalysts' structure. The incorporation of Ir single atoms caused the lattice expansion of the $\text{Cu}_{0.3}\text{Co}_{2.7}\text{O}_4$ due to the larger atomic radius of the Ir atoms than that of Co atoms, which led to the redshift of the characteristic peaks in the Raman spectra.

4. The authors claimed the obtained $\text{Ir}_1/\text{Cu}_{0.3}\text{Co}_{2.7}\text{O}_4$ with $d = 0.6$ nm exhibits excellent acidic OER catalytic stability. Catalyst stability should be compared to commercial IrO_2 stability toward acidic OER.

Response: We agree with the reviewer that our catalyst performance needs to be compared to commercial IrO_2 . We tested the acidic OER catalytic stability of commercial IrO_2 in 0.1 M HClO_4 electrolyte and added the results to Fig. 5d. The result showed that $\text{Ir}_1/\text{Cu}_{0.3}\text{Co}_{2.7}\text{O}_4$ with $d = 0.6$ nm exhibited better stability in comparison to that of commercial IrO_2 (Fig. R11d). The correlated discussions have been added to our manuscript.

Fig. R11 | Stability evaluation towards acidic oxygen evolution. **a**, Polarization curves of catalysts at different scan cycles in 0.1 M HClO_4 electrolyte. The displayed polarization curves are the 1, 200, 400, 600, 800, and 1000 cycles, respectively. **b**, Decrease in current densities under different scan cycles of $\text{Cu}_{0.3}\text{Co}_{2.7}\text{O}_4$ and $\text{Ir}_1/\text{Cu}_{0.3}\text{Co}_{2.7}\text{O}_4$ with different Ir-Ir distances. The selected overpotential was 1.60 V (E vs RHE) for all catalysts. **c**, Dissolution of Co species under different scan cycles of $\text{Cu}_{0.3}\text{Co}_{2.7}\text{O}_4$ and $\text{Ir}_1/\text{Cu}_{0.3}\text{Co}_{2.7}\text{O}_4$ with different Ir-Ir distances. **d**, Chronopotentiometry curves of $\text{Cu}_{0.3}\text{Co}_{2.7}\text{O}_4$, $\text{Ir}_1/\text{Cu}_{0.3}\text{Co}_{2.7}\text{O}_4$ with different Ir-Ir distances, $E = 1.60$ V, and commercial IrO_2 towards acidic OER at 10 mA cm^{-2} .

5. After the durability test, the lattice fringe of the $\text{Ir}_1/\text{Cu}_{0.3}\text{Co}_{2.7}\text{O}_4$ with $d = 0.6$ nm became ambiguous. Please explain the origin of the phenomenon.

Response: We genuinely thank the reviewer for the valuable suggestion. During the electrode preparation process, Nafion was used to bond the catalyst and electrode. After the stability test, $\text{Ir}_1/\text{Cu}_{0.3}\text{Co}_{2.7}\text{O}_4$ with $d = 0.6$ nm was isolated by centrifugation, the precipitates were washed several times with deionized water and ethanol. However, a trace amount of Nafion remained on the catalyst surface, thus affecting the imaging of the HAADF-STEM image. The degradation of imaging quality caused the lattice fringe of the $\text{Ir}_1/\text{Cu}_{0.3}\text{Co}_{2.7}\text{O}_4$ with $d = 0.6$ nm to become ambiguous.

6. After introducing the Ir single atoms with $d = 0.6$ nm to the $\text{Cu}_{0.3}\text{Co}_{2.7}\text{O}_4$, the $\text{Ir}_1/\text{Cu}_{0.3}\text{Co}_{2.7}\text{O}_4$ presented no obvious degradation over a 60-hour stability test for acidic OER. To further illustrate the excellent stability of this sample, please add the change in the applied voltage after stabilization to Fig. 5d.

Response: Thank this reviewer for the valuable advice to improve the quality of this work. We have added the change in the applied voltage after stabilization to Fig. 5d. After a 60-hour stability test, the applied voltage of the $\text{Ir}_1/\text{Cu}_{0.3}\text{Co}_{2.7}\text{O}_4$ with $d = 0.6$ nm slightly increased by 20 mV, demonstrating its excellent stability during acidic OER (Fig. R11d).

7. Did the catalytic performance of $\text{Ir}_1/\text{Cu}_{0.3}\text{Co}_{2.7}\text{O}_4$ with $d = 0.6$ nm toward acidic OER lead among the spinel oxides? It would be better for the authors to give a figure to compare the catalytic performance of $\text{Ir}_1/\text{Cu}_{0.3}\text{Co}_{2.7}\text{O}_4$ with $d = 0.6$ nm with some other spinel oxides.

Response: We genuinely thank the reviewer for the valuable suggestion. We have added the comparison of the catalytic performance for $\text{Ir}_1/\text{Cu}_{0.3}\text{Co}_{2.7}\text{O}_4$ with $d = 0.6$ nm with some other spinel oxides to our manuscript as Supplementary Figure 18 and Table 2. The overpotential at 10 mA cm^{-2} of $\text{Ir}_1/\text{Cu}_{0.3}\text{Co}_{2.7}\text{O}_4$ with $d = 0.6$ nm was on par with the best records of currently reported Co-based OER catalysts (Fig. R12 and Table R1).

Figure R12 | Electrocatalytic performance comparison. Comparison of overpotentials at a current density of 10 mA cm^{-2} for currently reported acidic OER catalysts.

Table R1. Comparison of oxygen evolution performance for recently reported Co-based catalysts in acidic electrolyte.

Catalysts	Electrolyte	Overpotential (mV) @ $j = 10 \text{ mA cm}^{-2}$	Ref.
$\text{Ir}_1/\text{Cu}_{0.3}\text{Co}_{2.7}\text{O}_4$ with $d = 0.6$ nm	0.1 M HClO_4	290	This work

$\text{Co}_{3-x}\text{Ba}_x\text{O}_4$	0.5 M H_2SO_4	278	R1
$\text{Ir}_{0.06}\text{Co}_{2.94}\text{O}_4$	1.0 M HClO_4	292	R2
$\text{Co}_3\text{O}_4@\text{C-GS}$	pH 1.0 H_2SO_4	350	R3
La,Mn-codoped Co_3O_4	0.1 M HClO_4	353	R4
$\text{Co}_3\text{O}_4@\text{C/GPO}$	1.0 M H_2SO_4	356	R5
$\text{Co}_3\text{O}_4@\text{C/CP}$	0.5 M H_2SO_4	370	R6
Co_2MnO_4	pH 1.0 H_2SO_4	395	R7
$\text{Co}_3\text{O}_4/\text{CeO}_2$	0.5 M H_2SO_4	423	R8
$\text{C}@\text{CeO}_2/\text{Co}_3\text{O}_4$	0.5 M H_2SO_4	425	R9

Reviewer #3 (Remarks to the Author):

Response: Thanks for your valuable comments, which significantly improved the quality of our manuscripts.

Revision References

- R1. Wang, N. et al. Doping shortens the metal/metal distance and promotes OH coverage in non-noble acidic oxygen evolution reaction catalysts. *J. Am. Chem. Soc.* **145**, 7829-7836 (2023).
- R2. Shan, J. et al. Short-range ordered iridium single atoms integrated into cobalt oxide spinel structure for highly efficient electrocatalytic water oxidation. *J. Am. Chem. Soc.* **143**, 5201-5211 (2021).
- R3. Liu, Z. et al. Interface engineering a high content of Co^{3+} sites on Co_3O_4 nanoparticles to boost acidic oxygen evolution. *Langmuir* **39**, 16415-16421 (2023).
- R4. Chong, L. et al. La- and Mn-doped cobalt spinel oxygen evolution catalyst for proton exchange membrane electrolysis. *Science* **380**, ade1499 (2023).
- R5. Yu, J. et al. Sustainable oxygen evolution electrocatalysis in aqueous 1 M H_2SO_4 with earth abundant nanostructured Co_3O_4 . *Nat. Commun.* **13**, 4341 (2022).
- R6. Yang, X. et al. Highly acid-durable carbon coated Co_3O_4 nanoarrays as efficient oxygen evolution electrocatalysts. *Nano Energy* **25**, 42-50 (2016).

- R7. Li, A. et al. Enhancing the stability of cobalt spinel oxide towards sustainable oxygen evolution in acid. *Nat. Catal.* **5**, 109-118 (2022).
- R8. Huang, J. et al. Modifying redox properties and local bonding of Co_3O_4 by CeO_2 enhances oxygen evolution catalysis in acid. *Nat. Commun.* **12**, 3036 (2021).
- R9. Liu, H. et al. Boosting $\text{CeO}_2/\text{Co}_3\text{O}_4$ heterojunctions acidic oxygen evolution via promoting OH coverage. *ACS Appl. Energy Mater.* **6**, 8949-8956 (2023).

REVIEWERS' COMMENTS

Reviewer #1 (Remarks to the Author):

The authors have answered/addressed all of our questions/comments, so the manuscript may be accepted for publication as is now.

Thank you

Reviewer #2 (Remarks to the Author):

The authors have answered all the questions and this manuscript can now be accepted in the present form.

Reviewer #3 (Remarks to the Author):
